# Ultra-High Contrast MRI: Using Divided Subtracted Inversion Recovery (dSIR) and Divided Echo Subtraction (dES) Sequences to Study the Brain and Musculoskeletal System

**DOI:** 10.3390/bioengineering11050441

**Published:** 2024-04-29

**Authors:** Daniel Cornfeld, Paul Condron, Gil Newburn, Josh McGeown, Miriam Scadeng, Mark Bydder, Mark Griffin, Geoffrey Handsfield, Meeghage Randika Perera, Tracy Melzer, Samantha Holdsworth, Eryn Kwon, Graeme Bydder

**Affiliations:** 1Mātai Medical Research Institute, Tairāwhiti Gisborne 4010, New Zealand; 2Department of Anatomy and Medical Imaging—Faculty of Medical and Health Sciences & Centre for Brain Research, University of Auckland, Auckland 1010, New Zealand; 3Te Whatu Ora Tairawhiti, Gisborne 4010, New Zealand; 4Insight Research Services Associated, Gold Coast 4215, Australia; 5Auckland Bioengineering Institute, University of Auckland, Auckland 1010, New Zealand; 6Department of Medicine, University of Otago, Christchurch 8011, New Zealand; 7New Zealand Brain Research Institute, Christchurch 8011, New Zealand; 8Department of Radiology, University of California, San Diego, CA 92093, USA

**Keywords:** MRI, contrast, traumatic brain injury, white matter, ultrashort-T_2_*, fascia, multiple sclerosis

## Abstract

Divided and subtracted MRI is a novel imaging processing technique, where the difference of two images is divided by their sum. When the sequence parameters are chosen properly, this results in images with a high T_1_ or T_2_ weighting over a small range of tissues with specific T_1_ and T_2_ values. In the T_1_ domain, we describe the implementation of the divided Subtracted Inversion Recovery Sequence (dSIR), which is used to image very small changes in T_1_ from normal in white matter. dSIR has shown widespread changes in otherwise normal-appearing white matter in patients suffering from mild traumatic brain injury (mTBI), substance abuse, and ischemic leukoencephalopathy. It can also be targeted to measure small changes in T_1_ from normal in other tissues. In the T_2_ domain, we describe the divided echo subtraction (dES) sequence that is used to image musculoskeletal tissues with a very short T_2_*. These tissues include fascia, tendons, and aponeuroses. In this manuscript, we explain how this contrast is generated, review how these techniques are used in our research, and discuss the current challenges and limitations of this technique.

## 1. Introduction: Soft Tissue Contrast

Soft tissue contrast is what differentiates different parts of the anatomy, and more importantly, normal from abnormal anatomy, on a medical image. The historic advantage of Magnetic Resonance Imaging (MRI) is that it provides superior soft tissue contrast compared to radiography, computed tomography, and ultrasound. The brightness, or signal, of a particular tissue on an MR image is based on specific properties of that tissue, combined with the particulars of the pulse sequence used for image acquisition. Contrast is the difference in signal between two tissues and/or fluids seen on an image.

Common tissue properties used for generating MR images include mobile proton density (ρ_m_), longitudinal relaxation (T_1_), and transverse relaxation (T_2_). These were first described by Felix Bloch in 1946 [1] but not in the context of medical imaging. Different tissues, by virtue of their different intra- and extra- cellular characteristics, have different tissue properties. Most clinical MR techniques are based on exploiting differences in tissue ρ_m_, T_1_, and T_2_. These include spin echo (SE), gradient echo, and inversion recovery (IR). 

Other tissue properties include mean diffusivity (D*), susceptibility, perfusion, vascular permeability, chemical shift, and velocity. More advanced MRI techniques take advantage of these properties to generate contrast. For example, diffusion-weighted imaging (DWI), contrast-enhanced imaging, and phase-contrast imaging generate contrast based on differences in D*, perfusion/vascular permeability, and velocity, respectively.

Some images are useful in their ability to distinguish different types of anatomy. Examples include the need to distinguish intracranial hemorrhage from normal gray and white matter, colon cancer metastases from the normal surrounding liver, and fluid from the ends of a torn and retracted muscle or tendon. In these examples, the pathology manifests as tissues with fundamentally different tissue properties. The ρ_m_, T_1_, and T_2_ of blood are sufficiently different from those of gray or white matter to differentiate these tissues on an anatomic image. Colon cancer metastases have a sufficiently different ρ_m_, T_1_, T_2_, and vascular permeability to liver to differentiate them from normal liver. Fluid and hemorrhage have a sufficiently different ρ_m_, T_1_, and T_2_ from those of muscle and tendon to visualize them on MR images. 

Some types of images are particularly valuable because of their ability to produce useful contrast from small changes in tissue properties from normal. Examples include diagnosing primary neoplasms and inflammation. Metastatic disease is relatively straightforward to diagnose, because the cancerous tissue is usually fundamentally different from the tissue it has metastasized into. However, primary neoplasms often have similar tissue properties to the organs they arise in. It is therefore difficult to identify hepatocellular carcinoma in the liver based only on ρ_m_, T_1_, and T_2_. For this reason, intravenous Gadolinium-enhanced MRI is used to generate contrast within the liver, based on differences in perfusion and capillary permeability. Likewise, the ρ_m_, T_1_, and T_2_ changes in gray and white due to acute stroke are also very small. Detection of acute stroke uses a technique that generates contrast based on changes in tissue D*.

The earliest MRI pulse sequences (or image acquisition techniques) were often better than CT in identifying normal anatomy: SE and gradient echo pulse sequences take advantage of differences in ρ_m_, T_1_, and T_2_ to provide excellent anatomic images of the brain, musculoskeletal system, and abdominal organs. Varying certain specifics of the image acquisition allows a sequence to be sensitive to differences in tissue ρ_m_, T_1_, or T_2_, leading to the concept of ρ_m_-, T_1_-, and T_2_-weighted images, where the brightness of a tissue on an image is mostly attributable to a single tissue property.

The evolution of MRI since the mid 1980’s has been to develop improved techniques for distinguishing normal anatomy from changes due to disease. Two examples include imaging in multiple sclerosis (MS) and imaging short T_2_* musculoskeletal tissues such as cortical bone, fascia, tendons, and aponeuroses.

### 1.1. Imaging of MS

In the early 1980s, IR techniques, which will be explained later in the manuscript, distinguished MS plaque from background white matter, based on the T_1_ differences between plaque and normal white matter [2]. Increased plaque T_1_ compared to white matter T_1_ resulted in an increased signal on the images. Subsequent SE techniques were able to identify plaque based on differences in T_2_ [3,4,5,6]. Increased plaque T_2_ compared to white matter T_2_ resulted in an increased signal on the images. 

Because plaques were noted to result in both increased white matter T_1_ and T_2_, the STIR sequence was developed in 1985 to generate increased contrast due to increases in both tissue T_1_ and T_2_ [7]. This improved sensitivity in detecting subtle plaques meant that white matter with smaller increases in T_1_ and T_2_ could be discerned from the normal background tissue. In 1992, the FLAIR sequence was developed to null bright signals from cerebrospinal fluid (CSF), making it easier to see changes near the extra-axial spaces [8]. This provided a visually appealing image, and FLAIR has become the standard-of-care clinical sequence for imaging cerebral white matter. However, with the FLAIR sequence, the effects of T_1_ and T_2_ on tissue brightness are opposed: increased tissue T_1_ results in a decreased signal and increased tissue T_2_ results in an increased signal. STIR may therefore be more sensitive for detecting subtle plaque, because the T_1_ and T_2_ effects on the signal are synergistic.

Techniques such as DIR (double IR) [9] and MP2RAGE [10], developed in 1985 and 2010, respectively, take two IR sequences obtained with different parameters and multiply them together to create images sensitive to small changes in T_1_ and T_2_. Both sequences have been shown to be superior for detecting focal MS plaques in the gray matter compared to the standard of care, FLAIR [11,12]. In 2019, the FLAWShc (Fluid and White matter Suppression, high contrast) sequence was described [13], in which two IR sequences with widely different inversion times are subtracted, then divided by their sum, similar to the divided Subtracted Inversion Recovery (dSIR) sequence described in this paper, but not using magnitude reconstruction or variable inversion times. FLAWShc sequences have been used to detect cortical lesions in MS [14]. DIR, MP2RAGE, and FLAWShc are noted for their high gray/white matter contrast.

Diffusion-weighted and intravenous Gadolinium-enhanced images have been used to image MS but have not proven to be more sensitive for detecting early changes in the disease. Rather, changes in plaque D* and perfusion/vascular permeability are used as markers of active inflammation at previously known sites of disease [15].

Although the imaging findings for MS are focal, MS is a systemic disease of the white matter [16]. Focal MS plaques represent permanent damage to the white matter and do not regress with treatment. We have hypothesized that subtle, potentially reversible, pathologic changes in white matter are manifest on imaging as small changes in T_1_ that are too small to detect with currently used sequences. We also hypothesize that these more subtle diffuse changes (which we are calling neuroinflammation, without knowledge of its true etiology) exist in other generalized brain disorders, such as mild traumatic brain injury (mTBI), substance abuse, and post-viral syndromes such as long COVID.

With this in mind, the subtracted inversion recovery (SIR) image was described in 2017 and 2018 as a pixel-by-pixel subtraction of two IR sequences, resulting in an image with an increased sensitivity to small changes in T_1_ but only between two specific values of T_1_ [17,18]. The range of T_1_s where the sequence is highly sensitive to changes is called the middle domain (mD) of the sequence. This allows for targeted imaging of the white matter that is twice as sensitive to small changes in white matter T_1_ than current clinical sequences. 

In 2022 and 2023 we described the dSIR technique, which is even more sensitive to small changes in T_1_ [19,20]. This sequence takes two images obtained using the IR technique and generates a third image by performing a voxel-by-voxel division of their difference by their sum. This is called “divided subtraction”. As we will demonstrate, dSIR images show high sensitivity in imaging small changes in white matter T_1_ from normal.

### 1.2. Soft Tissue Contrast in MSK

Soft tissue contrast in musculoskeletal (MSK) imaging presents unique challenges. Tissues with short (1–10 ms) and ultrashort (0.1–1 ms) T_2_*s are difficult to image, but a wide range of techniques have been developed to do this. The relevant tissues include cortical bone, myelin, fascia, aponeuroses, and lung. The T_2_* tissue property is similar to the T_2_ tissue property and, in the context of short TE imaging, will be used interchangeably. As we will show later, the signal from these tissues quickly drops to zero after excitation, making them difficult to image directly without specialized techniques.

Advances in gradient hardware and radiofrequency (rf) switching have facilitated the development of ultrashort TE (UTE) and zero TE (zTE) sequences that can capture signals from short and ultrashort T_2_* tissues. Longer T_2_ soft tissues (such as white matter, gray matter, fat, abdominal soft tissues, muscle, blood, and fluid) also appear bright on these images, providing that repetition times (TR—to be discussed subsequently) are chosen to allow full recovery of their longitudinal magnetizations. 

Imaging with zTE captures the signal from cortical bone and fascia, which still remain low-signal compared to other tissues due to their low mobile proton densities [21,22]. Long T_2_* tissues are high-signal on these images but show little contrast between them. Following bias correction and intensity normalization, these images can be displayed with an inverted gray scale, so that the cortical bone and fascia appear high-signal (bright) and long T_2_* tissues appear low-signal (dark). Most currently available commercial MR scanners include this option for imaging cortical bone (Figure 1).

An echo subtraction (ES) technique has been described, in which images obtained using a short TE are subtracted from images obtained with an ultrashort TE. The workings of this sequence are explained subsequently. ES is used to suppress signals from long T_2_* tissue components. In 2018, the 3D DIR-UTE-cones sequence was described [23], which uses an ultrashort TE to detect signals from ultrashort T_2_* tissues and suppresses signals from longer-T_2_* water- and fat-containing tissues by using adiabatic inversion pulses to null their longitudinal magnetizations.

In 2022, we introduced the divided echo subtraction (dES) sequence [24]. The dES sequence obtains two images with ultrashort and short echo times and performs a voxel-by-voxel subtraction, followed by division by the sum of the two images. ES produces a band-pass T_2_* filter with a low signal in the lower ultrashort T_2_* domain, and dES produces a low-pass T_2_* filter with a high signal in the lower ultrashort T_2_* domain. This is important for producing a high signal from cortical bone and other tissues such as myelin with significant low ultrashort T_2_* components.

## 2. Divided Subtraction MRI

The divided subtraction technique can be generalized as a simple mathematical manipulation of otherwise basic MRI pulse sequences, as will be explained subsequently. We have applied it to T_1_-weighted images for the targeted evaluation of white matter and separately to ultrashort T_2_-weighted images for the evaluation of bone and fascia. The T_1_ technique is called dSIR. The T_2_ technique is called dES.

Division and subtraction is a novel way of generating increased MR contrast from otherwise basic and readily available pulse sequences. The image processing is fast and comprised of simple image arithmetic. In addition to imaging white matter, dSIR can also produce images targeted to specific tissues and small changes in the T_1_ of these tissues due to disease. The technique can easily be optimized for different diseases in different parts of the body. We call this approach targeted MRI (tMRI). For T_2_-weighted images, dES is used to create images of cortical bone, fascia, aponeuroses, and tendons. 

This review introduces the divided subtracted technique, as implemented for T_1_- and T_2_-weighted images in the brain and musculoskeletal systems, respectively. We introduce the tissue property filter paradigm to explain how this contrast is generated. This paradigm simplifies the workings of many MR imaging techniques but is rarely used in the literature. We describe how we implemented divided subtraction sequences in our research and provide specific examples. We also discuss the current challenges and limitations of this technique, including comparisons with similar methods. Lastly, we provide examples of other potential applications.

## 3. Tissue Contrast in MRI and Theory behind dSIR and dES

When placed into a strong, static magnetic field (B_0_), the magnetic moments of a patient’s water protons align to form a net magnetization (M_0_) parallel to the B_0_. The M_0_ can be rotated by applying a radiofrequency (rf) pulse at the Larmour frequency for hydrogen at the field strength of the B_0_. Once rotated, the net magnetic moment precesses about the axis of the B_0_ and can be represented in the rotating frame by two components: a longitudinal magnetization (M_z_) parallel to the axis of the B_0_ and a transverse magnetization (M_xy_) perpendicular to the B_0_. 

The following Bloch equations describe the behavior of M_z_ and M_xy_ immediately after the M_0_ is rotated 90°, with the flip being instantaneous and occurring at t = 0. T_1_ and T_2_ are tissue properties specific to the tissue being imaged.
(1)$$M_{z}\left(t\right)=M_{0}\left(1-e^{-t/T1}\right)$$
(2)Mxyt=M0 e−tT2

At time t, a detector measures the magnitude of M_XZ_, which is referred to as signal S. An MR pulse sequence is a set of machine instructions to apply an rf pulse to the tissue magnetization, wait a predefined period, then measure this signal. This process is repeated multiple times. By varying the time between repetitive rf pulses (called TR) and the time between the rf pulse signal detection (called TE), the measured signal varies based on tissue ρ_m_, T_1_, T_2_, or some combination of all three. The TR and TE are referred to as the pulse sequence parameters and are under the control of the operator.

### 3.1. Spin Echo

Traditional SE pulse sequences use a flip angle (α) of 90° and a TR sufficiently long such that M_xy_ = 0 when t in Equation (2) equals TR. The signal S emanating from a specific tissue as measured by the detector is
(3)=K ρm×1−e−TR/T1×e−TET2
where K is a scaling function. A more explicit rendering of Equation (3) in terms of tissue properties is as follows:(4)$$S=S_{\rho_{m}}\left(\rho_{m}\right)\times S_{T_{1}}\left(T_{1}\right)\times S_{T_{2}}\left(T_{2}\right)$$
(5)$$S_{PD}\left(\rho_{m}\right)=\rho_{m}$$
(6)$$S_{T_{1}}\left(T_{1}\right)=\left(1-e^{-TR/T_{1}}\right)$$
(7)ST2T2=e−TET2

It should be noted that the fast spin echo (FSE) technique is almost uniformly used instead of the SE technique. Equations (3)–(7) are also valid for the FSE technique, with the caveat that the parameter TE actually refers to an “effective” TE. The specific differences between SE and FSE are beyond the scope of this manuscript, and the terms are used here interchangeably.

Equations (4)–(7) recast Equation (3) into three independent components each dependent on a tissue property that is explicitly identified as the dependent variable and a sequence parameter that is explicitly identified as a constant parameter of the sequence. We call Equations (5)–(7) tissue property filters: one for $\rho_{m}$, one for T_1_, and one for T_2_. They allow for a mathematical explanation of how changes in tissue property affect the signal. This paradigm was first described by Bydder and Young in 2020 [25].

Figure 2 shows the ρ_m_, T_1_, and T_2_ tissue property filters for the SE sequence with α = 90°, TR = 700 ms, and TE = 5 ms. Each plot shows the contribution of a tissue property to differences in signal, or contrast, between tissues. Most soft tissues have the same ρ_m_, and therefore ρ_m_ does not generate contrast. Most soft tissues are distributed along the steep part of the $S_{T_{1}}\left(T_{1}\right)$ curve, such that their different T_1_ properties result in contrast (i.e., a different signal). While some tissues have similar T_1_ values (white matter and liver, for example), these are in different parts of the body and are not typically imaged at the same time. Most tissues are distributed along the flat part of the $S_{T_{2}}\left(T_{2}\right)$ curve, such that their different T_2_ properties do not result in contrast. Muscle and tendons are an exception. This is therefore called a “T_1_-weighted” sequence, because *most* of the contrast between tissues is due to their different T_1_ values. In practice, all sequences produce a mixed contrast, and the relative amount of contrast due to each tissue property can be derived mathematically; however, this is beyond the scope of this discussion.

Figure 3 shows the $\rho_{m}$, T_1_, and T_2_ tissue property filters for the SE sequence with α = 90°, TR = 5000 ms, and TE = 100 ms. Most soft tissues have the same $\rho_{m}$, and therefore $\rho_{m}$ does not generate contrast. Most soft tissues are distributed along the flat part of the $S_{T_{1}}\left(T_{1}\right)$ curve, such that their T_1_ properties result in a similar signal. Most tissues are distributed along the steep part of the $S_{T_{2}}\left(T_{2}\right)$ curve, such their differences in T_2_ result in contrast. This is therefore called a “T_2_-weighted” sequence because *most* of the contrast between tissues is due to their different T_2_ values. 

T_1_- and T_2_-“weighted” SE and FSE sequences are generally good for distinguishing anatomy. However, as discussed previously, the detection of subtle disease requires small changes in a tissue property to result in large changes in signal. The sensitivity of a sequence for detecting these small changes within a tissue is the slope of the filter at the T_1_ or T_2_ value of the tissue. The slope, by definition, is the change in signal for a given change in tissue property. If the slope is positive, an increase in tissue property results in an increased signal. If the slope is negative, an increase in tissue property results in a decreased signal. For small changes in white matter T_1_ to result in a noticeable change in signal, the T_1_ of white matter must sit on a steep part of the $S_{T_{1}}\left(T_{1}\right)$ curve. The slope of Equation (6), with TR = 700 (Figure 2b) at the T_1_ of white matter (850 ms in a 3 Tesla scanner), is −4.3 × 10^−4^/ms (arbitrary signal units on the y-axis per ms on the x-axis). A typical “T_1_ weighed” FSE image of the brain is shown in Figure 4. Gray matter is darker than white matter, and fluid is darker than gray matter.

### 3.2. Inversion Recovery

An IR sequence uses a series of sequential 180° and 90° flips. The time between two 180° flips is called TR. The time between the 180° flip and 90° flip is called TI. The detector is turned on at a time TE, following the 90° flip. The equation that describes the behavior of M_Z_, and hence the signal due to T_1_ immediately after a 90° flip, is
(8)$$S\left(T_{1}\right)=\left|1-2e^{-\frac{TI}{T_{1}}}+e^{-\frac{TR}{T_{1}}}\right|.$$
and it is plotted in Figure 5a.

The slope of Equation (8), with TR = 5000 ms and TI = 1000 ms at the T_1_ of white matter, is −8.2 × 10^−4^/ms. The sensitivity of the IR sequence to small changes in the T_1_ of white matter is therefore nearly twice that of the “T_1_-weighted” SE sequence (Table 1). A typical “T_1_-weighted” FSE-IR image of the brain is shown in Figure 5b.

A specific feature of the IR sequence is that the signal is nulled for tissues with a specific T_1_. The T_1_ resulting in zero signal is determined by the choice of TI, or time between the 180° and 90° rf pulses. For a sufficiently long TR, the T_1_ that is nulled can be calculated by setting Equation (8) to zero and solving for T_1_. This results in T_1nulled_ = 1.45 × TI. 

### 3.3. Subtracted Inversion Recovery

Figure 6a shows the plot of two IR T_1_ tissue filters. The red curve is a plot of Equation (8), with TI = 580 ms (call this TI_short_) and TR = 5000 ms, and results in tissues with the T_1_ values of white matter being nulled. This results in an unusual image, where white matter is black and gray matter is bright (Figure 6b). Increases in white matter T_1_ from normal result in an increased signal, which is the opposite of what is expected from a standard T_1_-weighted sequence.

The blue curve in Figure 6a is a plot of Equation (8), with TI = 970 ms (call this TI_high_) and TE = 5000 ms, and results in tissues with the T_1_ values of gray matter being nulled. This results in an image like that produced by the IR T_1_ in Figure 5a, only the gray matter appears black.

The green curve in Figure 6a is the T_1_ filter for the SIR image and is the result of subtracting the blue curve from the red curve in the central region. Visually, the slope of the green curve is steeper than either the red or blue curves. The slope of the green curve at the T_1_ of the white matter (850 ms) is 16 × 10^−4^/ms, which is steeper than each of the individual T_1_ filters. On an SIR image, normal white matter is black, and small increases in white matter T_1_ result in an increased signal (Figure 7) This increased sensitivity for small changes in T_1_ is only applicable in the middle domain (mD), between the T_1_ values of white and gray matter. If TI_high_ is decreased, then the mD narrows, and the slope in the mD (and as a result, the contrast generated from a change in T_1_) is increased. Conversely, as the mD is widened, the contrast decreases.

### 3.4. Subtracted Division Inversion Recovery

The red and blue plots in Figure 8a are the same IR T_1_ tissue filters shown in Figure 6a. The purple plot is the T_1_ filter for the divided Subtracted Inversion Recovery (dSIR) image. This T_1_ filter is the difference of the blue and red graphs divided by their sum. The slope of this filter at the T_1_ of white matter is 43 × 10^−4^/ms. This is 2.6 times the slope of the SIR sequence, 5.4 times the slope of the IR sequence, and 10 times the slope of the SE sequence (Table 1). This increased sensitivity is only applicable for tissues with T_1_ values in the mD between the T_1_s nulled by the two IR acquisitions. As the mD narrows, the slope steepens. Figure 8b shows the bipolar dSIR T_1_ filter, where the blue plot is derived from Equation (8) with a TI_high_ of 750 ms. Compared to Figure 8a, the mD of the sDIR T_1_ filter is narrower and the slope within the mD is greater. The slope of this filter at the T_1_ of white matter is 87 × 10^−4^/ms, which is twenty times the slope of the SE filter in Figure 2b (Table 1). Normal white matter appears black on this image. Very small increases in white matter T_1_ from normal appear bright. 

Modifying TI_high_ and TI_short_ controls two aspects of the image. First, TI_long_ and TI_short_ determine what tissues are best imaged. Figure 8a,b show a dSIR filter targeted to detect small changes in white matter. Figure 8c shows a dSIR filter targeted to detect small changes in the T_1_ of gray matter. The red curve is Equation (8), with TI_short_ = 970 ms designed to null signal from gray matter. The blue curve is Equation (8), with TI_long_ = 1200 ms which was chosen to be higher than 970 ms. TI_long_ and TI_short_ can be chosen to target T_1_ changes within any range of tissues. Second, as ΔTI (TI_long_ − TI_short_) decreases, the slope of the filter in the mD, and thus the sensitivity to small changes in T_1_, increases.

When dSIR is targeted to white matter, the normal white matter is predominantly dark (Figure 9). When using a wide mD with TI_long_ targeted to null gray matter, the gray matter is bright. Using a narrow mD with TI_long_ targeted to null tissues with a T_1_ value between those of gray and white matter, the gray matter is an intermediate signal, and there is a high signal boundary between the white and gray matter and between white matter and CSF. The high signal boundaries can be explained mathematically [19] but are summarized as follows. Voxels located in the transition region at the boundary between two tissues contain mixtures of the two tissues and have T_1_ values between those of voxels which contain only one of the two tissues. If the T_1_ filter has a maximum value at a T_1_ between the T_1_ s of the pure tissues, the boundary will be high-signal. If the T_1_ filter has a minimum value between these two T_1_ values, the boundary will be low-signal. The literature describes at least 10 different types of white matter, each having different T_1_ values. These can be discerned on narrow mD dSIR images. The white matter is not homogeneously dark.

In summary, dividing the difference of two IR sequences (identical except for different inversion times) by their sum results in a sequence that is very heavily and almost purely T_1_ weighted for tissues with T_1_ values between those of the tissues nulled by the two inversion times. This is accomplished by acquiring two IR images, then performing a pixel-wise addition, subtraction, and division of the images. The dSIR sequence with a narrow mD produces an image that has, in theory, 20 times the T_1_ contrast compared to a T_1_ FSE sequence.

### 3.5. Echo Subtraction (ES) and Divided Echo Subtraction (dES)

Figure 10a shows the T_2_* filters for an ultrashort TE sequence with TE = 0.03 ms (red plot) and a short TE sequence with TE = 2.2 ms (blue plot). Cortical bone has a T_2_* of 0.3 ms and fascia has a T_2_* of 1–2 ms. The green curve in Figure 10b is the subtraction of the blue curve from the red curve and is the ES T_2_* filter. The ES image shows a low to intermediate signal from cortical bone but a high signal from tissues with a T_2_* similar to fascia. This provides a contrast with longer T_2_* tissues whose signal is suppressed.

The purple curve in Figure 10c is the division of the difference of the red and blue curves by their sum and is the dES T_2_* filter. In the low ultrashort T_2_* range of about zero to 0.3 ms, the signal from the dES filter is much higher than that from the ES filter. 

## 4. Clinical Experience with dSIR

Our experience with dSIR is limited to case reports and small case–control studies, but the results are striking, and we believe there is potential for this technique to change how white matter diseases are imaged. 

It was initially noticed that dSIR identifies multiple sclerosis (MS) plaques not seen on conventional T_2_-FLAIR images (Figure 11), despite T_2_-FLAIR being a principal part of the gold standard for such imaging. Most pathology in white matter, including edema, ischemia, and demyelinating plaque, results in increases in white matter ρ_m_, T_1_, and T_2_. In this instance, the increased T_2_ in the plaque is insufficient to result in contrast on the T_2_-FLAIR images, but the small increase in white matter T_1_ is detected on dSIR images. It was originally thought that dSIR could be used to detect plaques occult on T_2_-FLAIR. This might be useful, because the typical imaging findings in MS are focal, and disease progression is monitored by serial evaluation of the size and characteristics of plaques. Unfortunately, the presence of high signal boundaries makes this difficult. Plaques can masquerade as gray matter, and the immediately subcortical regions are difficult to visualize (Figure 12). It is also not obvious that the ability to simply identify more white matter plaques would make a difference in how MS is diagnosed, staged, and followed.

However, MS is a systemic disease of the white matter. Imaging early and potentially reversible changes in the white matter using dSIR could potentially change the way MS is imaged. In one MS patient having an acute decompensation, the narrow mD dSIR images show geographic, as opposed to focal, changes in the white matter (Figure 13). We do not have a longitudinal follow-up, nor do we have sufficient patients at our institution to perform a longitudinal study. However, collaborators at larger institutions are planning to study this. 

We believe that dSIR is well suited to identifying generalized changes in white matter T_1_. We have described the “white out sign” as dSIR findings of a widespread and relatively homogeneous high signal throughout the white matter (Figure 14). This contrasts with normal, predominantly low-signal white matter. A third category of a widespread, less increased signal represents an intermediate form of the white out sign.

We have used dSIR to identify acute and chronic generalized white matter changes otherwise occult on clinical imaging. In two patients with suspected delayed post-hypoxic ischemic leukoencephalopathy (Grinker’s myelinopathy), dSIR demonstrated widespread increased white matter T_1_, despite a normal appearance of the white matter on T_2_-FLAIR images (Figure 15 and Figure 16) [26]. Both patients had experienced prolonged hypoxia due to attempted suicide. One was due to a drug overdose, and the other was due to asphyxia. Scans were obtained nine months and two years following injury. Grinker’s has been considered a rare disease, because conventional imaging is typically normal. The disorder may not be as rare as originally thought. dSIR could show obvious findings in symptomatic patients with previously normal conventional T_2_-FLAIR images.

In another example, a matched pair of scans in two teenage boys with impact-related head injury in the same rugby match showed widespread white matter changes in the symptomatic player but not in the asymptomatic one (Figure 17).

In a case–control cohort currently being prepared for journal submission, 27 of 33 patients with chronic post-concussive symptoms, clinically selected for being at high risk of having a brain abnormality, had a “white out” sign on dSIR images. The remaining five had intermediate findings. The T_2_-FLAIR showed no diffuse abnormalities in all 33 patients. An unmatched cohort of normal volunteers had normal white matter. Case–control studies do not prove correlation, and we are in the process of organizing a prospective study on acute mTBI patients to correlate the white matter changes seen on dSIR with the clinical course of symptoms.

Particularly encouraging, however, is anecdotal evidence that generalized white matter changes detected with dSIR sequences are reversible. Imaging in a patient following chronic methamphetamine use showed widespread white matter changes, which resolved following a prolonged period of abstinence (Figure 18). A similar reversibility was seen in a symptomatic patient following an mTBI (Figure 19). Imaging in the days following injury showed diffuse white matter changes which, along with the symptoms, resolved within two weeks. 

If dSIR can detect acute reversible changes in patients having acute MS flares, then it could be used as an imaging biomarker for treatment response. If reversible white matter changes consistently correlate with symptoms in mTBI, then imaging could be used to identify patients suffering damage from head trauma and possibly predict early recovery. In addition, demonstrating chronic white matter changes in patients with chronic symptoms but negative imaging provides an objective validation of disease. This is empowering for patients who have been told there is “nothing wrong with them”. It also provides a basis for funding treatment. Prospective trials need to be performed, but preliminary results suggest there is promise for using dSIR to detect subtle changes in white matter following insults to the brain. There is also potential for dSIR to play a similar role in post-viral syndromes such as long COVID.

dSIR is not limited to detecting changes in white matter. TI_short_ and TI_long_ can be chosen to maximize sensitivity to changes in tissue T_1_ between any two T_1_ values—hence, the term targeted MRI (tMRI). In Figure 8c, TI_short_ and TI_long_ are chosen to show small changes in gray matter T_1_. It would be interesting to compare this sequence with MP2RAGE and FLAWS hc for detecting cortical plaques in MS. MR fingerprinting has suggested that the primary difference between normal prostate tissue and prostate cancer is a decrease in tissue T_1_ [27]. However, differences in T_1_ between normal and cancerous tissues in the prostate are insufficient to produce contrast on standard FSE or IR T_1_-weighted FSE sequences. If TI_long_ is chosen to null normal peripheral zone prostate tissue and TI_short_ is chosen to null tissue with a shorter T_1_, normal peripheral zone tissue will appear bright and prostate cancers should appear dark. A proof of concept has been performed in normal volunteers [28]; however it is unknown if this approach will add specificity to the current techniques, which rely instead on T_2_ and D* for differentiating prostate cancer from normal background tissue. 

dSIR images are created from the pixel-by-pixel division of the difference of two standard IR images by their sum. These types of IR sequences are available on most clinical MR systems. The straightforward image arithmetic is easily performed using a MATLAB script. Technical barriers for implementation are therefore low.

## 5. Practical Limitations of dSIR and Future Technical Modifications

As implemented above, the dSIR sequence is not a high-resolution technique. The examples shown in Section 4 of the manuscript used a 2D fast spin echo technique, with 4 mm slices and an in-plane resolution of 1 × 1 mm. This is appropriate for detecting wide-spread changes in the white matter but may be limited for detecting small, focal changes, especially in the subcortical regions, where there is a high signal boundary between white and gray matter. The technique can also be performed using 3D high-resolution-magnetization-prepared gradient echo inversion recovery sequences, which can produce images with an isotropic 0.7 mm (or less) resolution. This is preferable for reducing artifacts due to volume averaging or when looking for focal abnormalities. The gradient echo readout modifies Equation 8, and the choices of TI_short_ and TI_long_ require mild modification.

There are also inherent limitations when performing analyses on images obtained at two separate time points. Even subtle patient motion between the two IR acquisitions will result in misregistration during the image arithmetic, resulting in artifacts on the dSIR images. Motion correction techniques such as radial/propeller/blade k-space trajectories in the data space, combined with rigid image registration techniques in the image space, can be used to correct for small amounts of motion prior to subtraction and division. These extra steps may limit the ability of older MR systems to create dSIR images. Our experience is that misregistration is uncommon in most patients.

dSIR images are also not intended to be viewed in isolation. The limited range of tissue T_1_s where there is image contrast means that large changes in tissue T_1_, or changes outside the mD, are missed. We therefore always review the dSIR images beside conventional T_2_-FLAIR and T_2_ images. This is especially important when looking for focal, as opposed to widespread, changes.

dSIR is inherently a targeted technique. The range of evaluated tissues is defined by the user through the choice of TI_short_ and TI_long_. This requires accurate knowledge of the T_1_ values of the tissues of interest, which is not always the case. If the TI_short_ chosen is too high then small changes in normal white matter T_1_ can result in a decreased, not increased, signal (Figure 20). The width of the mD is also critically important. If the mD is too wide, then the slope of the T_1_ bipolar filter in the mD may be insufficient to show subtle changes in T_1_. If the pathologic changes in T_1_ are larger than the width of the mD, then the findings can be less obvious, because the slopes of the dSIR T_1_ filters in Figure 8a,b become negative above the maximum T_1_ of the mD, which results in less increase in signal and lower contrast.

There are at least two approaches to targeting dSIR sequences. The first is to implement dSIR with a specific mD that accommodates variations in the normal T_1_ of white matter and allows for intermediate, as well as small, increases in white matter T_1_. This is our current approach with dSIR imaging. The TR is chosen long compared to the T_1_ of the tissues, typically around 5000 ms. When dividing the difference of the images by their sum, the T_2_ and $\rho_{m}$ filters cancel, so the choice of TE in theory is arbitrary, though we choose a low TE =7 ms to definitely remove T_2_ weighting from the process. Initial values for TI_short_ and TI_long_ were chosen in an attempt to accurately null white and gray matter, respectively, based on both the published values of gray and white matter T_1_ at 3T and also our experience using various inversion times on normal volunteers. However, we found that attempting to accurately null white matter resulted in many scans where the TI_short_ was a little too long, and the multiple attempts at accurately identifying TI_short_ resulted in long examination times. We ultimately chose TI_short_ = 350 ms and TI_long_ = 500 ms for studying generalized disease in the white matter. TI_short_ = 350 ms is below the value needed to null normal white matter but ensures that any increases in white matter T_1_ from normal will result in an increased signal. This results in normal white matter often having a mildly increased signal. The mD is narrow enough to ensure the steep slope of the T_1_ filter in the mD but wide enough to allow for a range of white matter increases in T_1_. The advantage of this approach is that useful dSIR imaging can be made available on virtually all clinical MRI scanners using existing IR sequences, without requiring close attention to sequence parameters. The scan time is short (both 2D and 3D data sets can be acquired in approximately five minutes), and the post-processing involves simple image arithmetic.

The second approach is to obtain dSIR images with a wide mD and mathematically create narrower mD images. The dSIR filter is approximately linear between the T_1_ values of the tissues nulled by TI_short_ and TI_long_ (Figure 8a,b and Figure 20). Within the mD, the signal of the T_1_ filter is approximated by the equation
(9)$$S_{dSIR}=\frac{ln4}{{TI}_{long}-{TI}_{short}}\times T_{1}-\frac{{TI}_{long}+{TI}_{short}}{{TI}_{long}-{TI}_{short}}$$

Solving for T_1_ turns the dSIR image into a T_1_ map, which can be used in Equation (8) to mathematically create images for any TI_long_ and TI_short_ that fall within the mD of the original dSIR mage. A synthetically created narrow mD dSIR image can then be computed. If TI_short_ and TI_long_ are initially chosen to encompass gray and white matter, then synthetic images can be optimized to the T_1_ values for gray and white matter for each patient. Unfortunately, this has proved difficult to accomplish, because it is not clear how to identify the T_1_ of “normal” white matter in cases of widespread disease.

The ability to create T_1_ maps from the dSIR images suggests two additional limitations of the technique. First, if dSIR images can be created mathematically from T_1_ maps, then why bother to obtain the two IR sequences if a T_1_ map is already available? This is a valid criticism. Standard IR sequences are readily available on most clinical MRI systems, but high-resolution T_1_-mapping techniques are not. The MP2RAGE sequence creates a T_1_ map encompassing all tissues on the image and is becoming increasingly common on clinical systems. The FLAWShc sequence can also be used to create a T_1_ map using a very wide mD. (FLAWShc could be considered a very wide mD dSIR technique but does not use magnitude reconstruction and does not produce a bipolar T_1_ filter). T_1_ maps created from both these sequences could be used to synthetically create subsequent images targeted at specific tissues. However, we believe that the more closely a range of T_1_ values are targeted by the original acquisition, the more accurate the subsequent T_1_ map will be. When looking for small changes in T_1,_ we want to use the most accurate technique available. Starting with a smaller, targeted mD may result in more accurate T_1_ maps for the tissues we want to evaluate. A comparison of these different T_1_-mapping techniques for the creation of dSIR images will need to be performed.

The second limitation is that the information contained on the dSIR image is essentially a linear mapping of T_1_ values to pixel brightness. If a simple T_1_ map of the tissues in the mD were available, then an equivalent image could be generated by creating an image lookup table having the same shape as the bipolar T_1_ filter in Figure 8a,b. dSIR encodes the contrast into the image data, which can be shared and viewed on any image viewer. The lowest signal corresponds to the tissue nulled by TI_short_, and the highest signal corresponds to the tissue nulled by TI_long_. It is straightforward to view the images using a full-dynamic-range linear gray scale lookup table which is available on all image viewers (specifically DICOM image viewers for medical images). It is not straightforward to design a bipolar, gray scale lookup table and import it into a viewer of choice. From this perspective, generating the dSIR image (either directly or synthetically) and encoding the contrast in the image file is more practical.

An additional limitation of the dSIR sequence as implemented for white matter is that the etiology of the small changes in T_1_ is unknown and may include other pathological processes besides neuroinflammation, including edema, demyelination, and degeneration. We have hypothesized that subtle neuroinflammation results in T_1_ increases too small to detect with conventional FSE and IR sequences. Whether this is accompanied by small increases in T_2_ is unknown, but frequently in disease, increases in T_1_ are accompanied by increases in T_2_. We are currently seeking funding to test dSIR in an animal model with histology as a reference standard. If dSIR is a marker of subtle neuroinflammation, then there is potential to use dSIR to triage mTBI patients who would benefit from anti-inflammatory therapies.

## 6. Future Validation

dSIR is an emerging technique and, despite the preliminary evidence presented above, has not undergone clinical validation. Prospective studies on symptomatic patients need to be performed to establish a correlation between symptoms, recovery, and white matter changes. This needs to be done separately for diseases like acute and chronic mTBI, MS, long COVID, and other post-viral syndromes. Histologic correlations need to be performed to determine the specific etiology of white matter changes. Prospective studies comparing dSIR to imaging techniques in other parts of the body also need to be performed. Examples include comparing dSIR to MP2RAGE and FLAWS hc for detecting cortical plaques in MS, and comparing dSIR to high b-value diffusion-weighted images for detecting cancers in the peripheral zone of the prostate.

## 7. Research Experience with dES

Musculoskeletal MRI as a field has primarily concerned itself with skeletal, joint, and muscular imaging. With improved imaging, we can advance our knowledge of tendon and fascial pathologies [28,29,30]. We have used dES to generate high-spatial-resolution images of the upper and lower extremities, with a high contrast between tendons/fascia/aponeurosis and the surrounding soft tissues. These images are incorporated into pipelines for creating finite-element models of the musculoskeletal system [31,32] which can simulate anatomical function [33,34]. dES and subtraction techniques using UTE sequences allow the novel visualization of fascia in vivo to understand anatomical variation in populations, explore its potential role in disease, and build realistic models to explore its role in musculoskeletal phenomena that are poorly understood [35].

On an ultrashort TE image acquired with TE = 0.03 ms, the cortical bone and fascia are dark and longer T_2_* tissues are bright (Figure 21a). Inverting the gray scale creates a “CT”-like image of the cortical bone (Figure 21b, also see Figure 1). Aponeuroses are also bright, but contrast with the surrounding soft tissues is low. The facial planes are not well delineated. An ES image created by subtracting a TE = 2.2 ms image from a TE = 0.03 ms image (Figure 21c) shows increased contrast between the connective tissues and adjacent soft tissues due to increased suppression of the higher T_2_ tissues, which appear darker than in Figure 21b. The facial planes are better delineated than on the inverted images. Cortex is not bright as predicted by the green curve in Figure 10b. A dES image created from TE = 0.03 ms and TE = 2.2 ms images shows increased contrast between the ultrashort T_2_ tissues and the background tissues. Fascial planes and aponeuroses are well visualized.

Imaging fascia in the extremities is technically challenging. There are at least three different tissue types in close proximity (short T_2_ fascia/aponeuroses, fat—which contains both long and short T_2_ components, and skeletal muscle). This type of imaging is prone to chemical shift artifacts and remains a work in progress. However, it serves as a practical example of using division and subtraction to increase soft tissue contrast.

## 8. Summary

Division and subtraction is a novel way of generating ultrahigh T_1_ and T_2_ contrast from otherwise basic and readily available MRI pulse sequences. The dSIR sequence produces T_1_-weighted images that are, in theory, 20 times more sensitive than traditional SE T_1_-weighted images for detecting small changes in T_1_ from normal. We currently use dSIR to identify widespread changes in white matter T_1_ in mTBI, MS, and other diffuse brain disorders. To our knowledge, no other conventional or advanced MRI technique can do this. Because dSIR can be targeted to be sensitive to T_1_ changes from normal within any tissue, we also plan to explore its utility in imaging prostate cancer. The dES sequence produces a similarly increased T_2_-weighted contrast for imaging short T_2_ tissues.

## Figures and Tables

**Figure 1 bioengineering-11-00441-f001:**
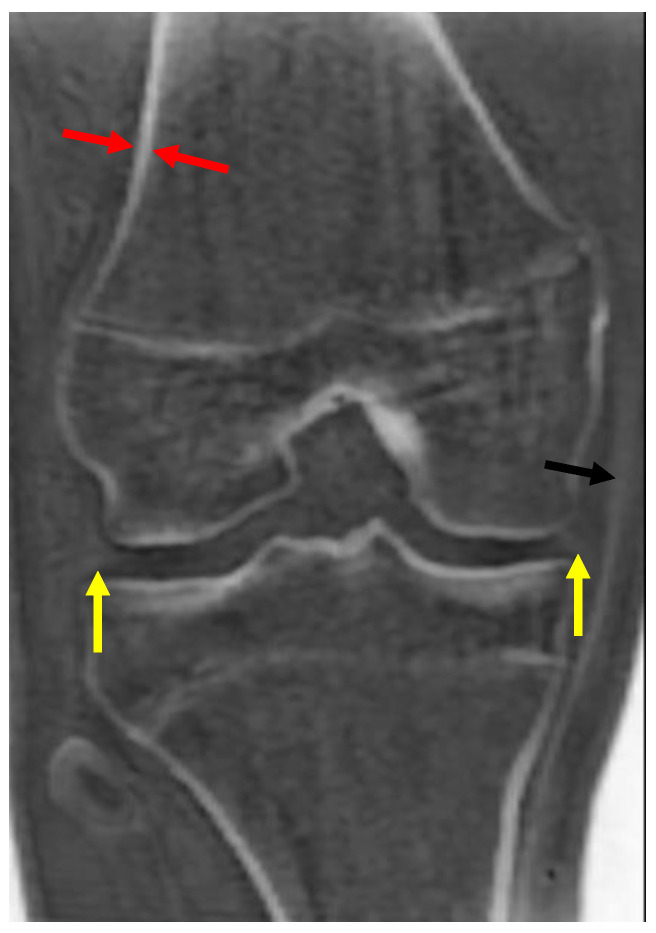
Coronal zTE image of the knee displayed with an inverted gray scale. Cortical bone (red arrow) is bright. Other short T_2_ tissues, such as the medial collateral ligament (black arrow) and the menisci (yellow arrows), are also bright, but less so than the cortical bone.

**Figure 2 bioengineering-11-00441-f002:**
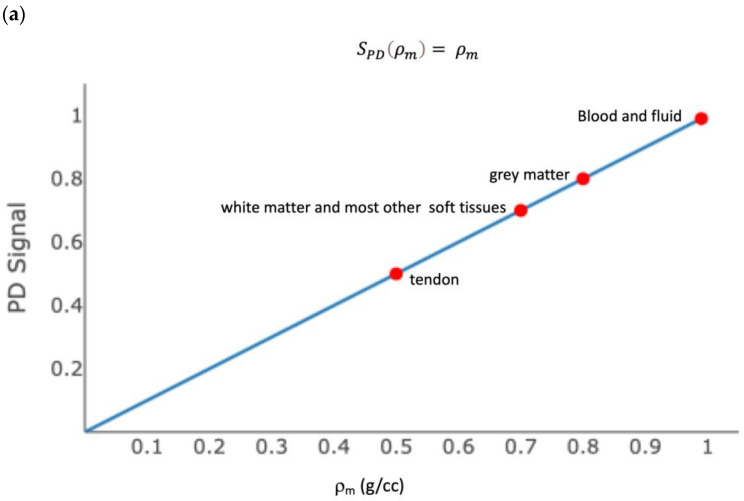
Tissue property filters for a T_1_-weighted image. (**a**) The proton density ($\rho_{m}$) filter for the fast spin echo sequence shows the signal due to r_m_ in an image. (**b**) The T_1_ filter for the fast spin echo sequence, with TR = 700 ms, shows the signal due to T_1_ in an image. Most tissues sit on the steep part of the curve, which results in different signals from different tissues based on T_1_ weighting. (**c**) The T_2_ filter for the fast spin echo sequence, with TE = 10 ms, shows the signal due to T_2_ in an image. Most tissues sit on the flat part of the curve, which results in little contrast between tissues based on T_2_ weighting. Muscle and tendon are the exception. Changes in tendon T_2_ are easily visualized on fast spin echo sequences, with TE = 10 ms, because the curve is steep at the T_2_ of tendon.

**Figure 3 bioengineering-11-00441-f003:**
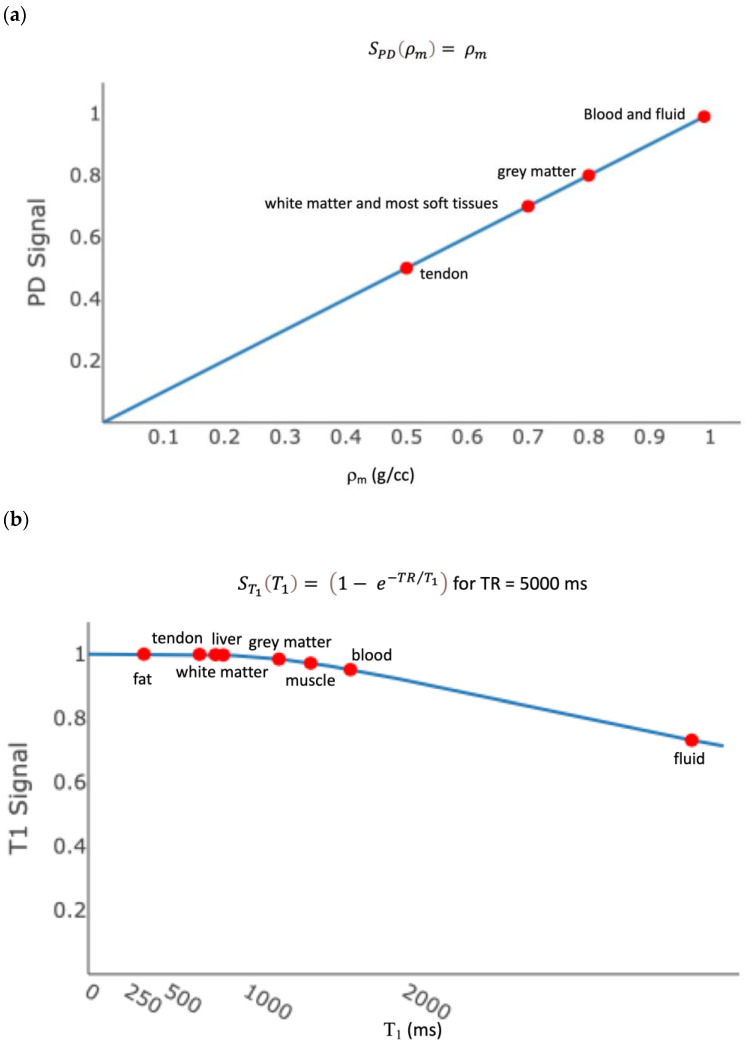
Tissue property filters for a T_2_-weighted image. (**a**) The proton density ($\rho_{m}$) filter for the fast spin echo sequence shows the signal due to r_m_ in an image. (**b**) The T_1_ filter for the fast spin echo sequence, with TR = 5000 ms, shows the signal due to T_1_ in an image. Most tissues sit on the flat part of the curve, which results in little signal difference between different tissues. (**c**) The T_2_ filter for the fast spin echo sequence, with TE = 100 ms, shows the signal due to T_2_ in an image. Most tissues sit on the steep part of the curve, which results in a contrast between tissues based on T_2_ weighting.

**Figure 4 bioengineering-11-00441-f004:**
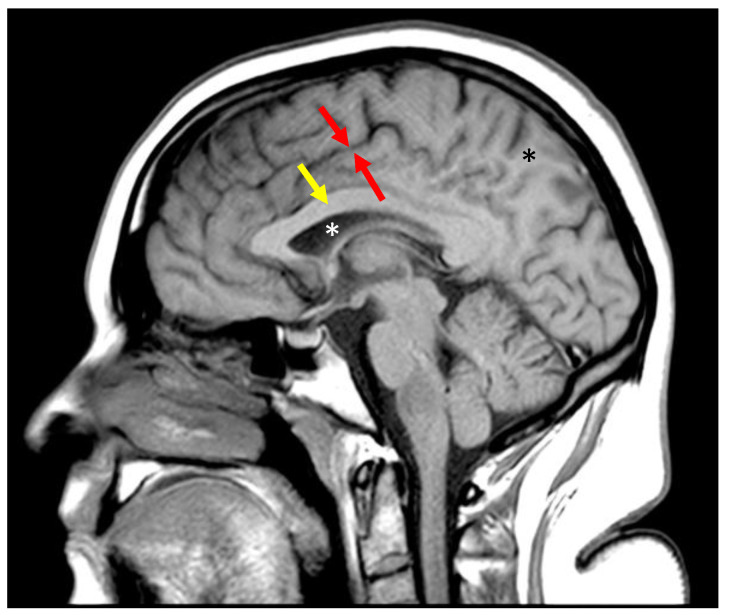
Sagittal fast spin echo image of the brain with TR = 700 ms and TE = 10 ms. Gray matter (between the red arrows) is darker than the subcortical white matter (black asterix) and corpus callosum (yellow arrow). Fluid (white asterix) is even darker. This parallels the curve in Figure 2b.

**Figure 5 bioengineering-11-00441-f005:**
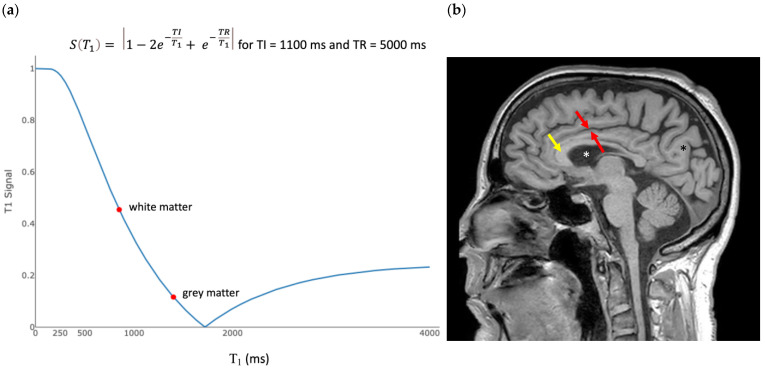
Inversion recovery T_1_ filter. (**a**) T_1_ tissue property filter for the IR sequence with TI = 1100 ms and TR = 5000 ms. Tissues with T_1_ = 1594 ms are nulled by these parameters. The slope of the curve in the region of most tissues of interest is negative, so that an increase in T_1_ results in a decreased signal. This produces an image like an SE T_1_-weighted image. The slope of the left half of the filter is steeper than the slope of the plot in Figure 2b, resulting in increased contrast. (**b**) Sagittal fast spin echo image of the brain with TR = 5000 ms and TE = 1000 ms. Gray matter (between the red arrows) is darker than the subcortical white matter (black asterix) and corpus callosum (yellow arrow). Fluid (white asterix) is even darker. This parallels the curve in (**a**). There is increased contrast compared to Figure 4, which reflects the larger difference in signal between gray and white matter in (**a**) than in Figure 2b.

**Figure 6 bioengineering-11-00441-f006:**
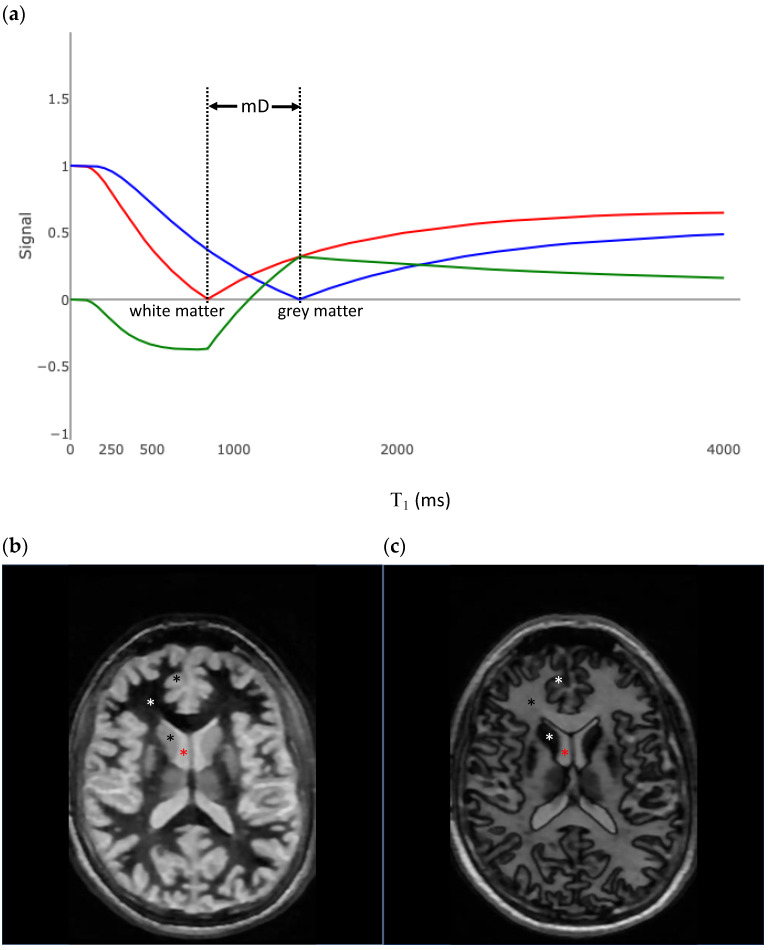
Subtracted inversion recovery (SIR) sequence. (**a**) T_1_ tissue property filter for the SIR sequence. The red curve uses a TI_short_ designed to null white matter. The blue curve uses a TI_long_ designed to null gray matter. The middle domain (mD) is the range of T_1_s between the tissues nulled by TI_short_ and Ti_long_. The green curve is the tissue filter for the SIR sequence and is the blue curve subtracted from the red curve. The slope of the green curve at the T_1_ of white matter is nearly two times the maximum slope of the red or blue curves. (**b**) Axial fast spin echo inversion recovery image, with TR = 5000 ms and TI = 580 ms, designed to null white matter. The slope of the T_1_ filter to the right of white matter (red curve in (**a**)) is reversed compared to the filters in 5a and 2b. Increases in T_1_ result in increased signal, and gray matter (black asterix) is brighter than white matter (white asterix). Fluid (red asterix) is brighter than gray matter. (**c**) Axial fast spin echo inversion recovery image, with TR = 5000 ms and TI = 970 ms, designed to null gray matter. The slope of the T_1_ filter to the right of the white matter (blue curve in (**a**)) is the same compared to the filters in Figure 5a and Figure 2b. Increases in T_1_ result in decreased signal, and gray matter (black asterix) is darker than white matter (white asterix). Fluid (red asterix) is brighter than gray matter.

**Figure 7 bioengineering-11-00441-f007:**
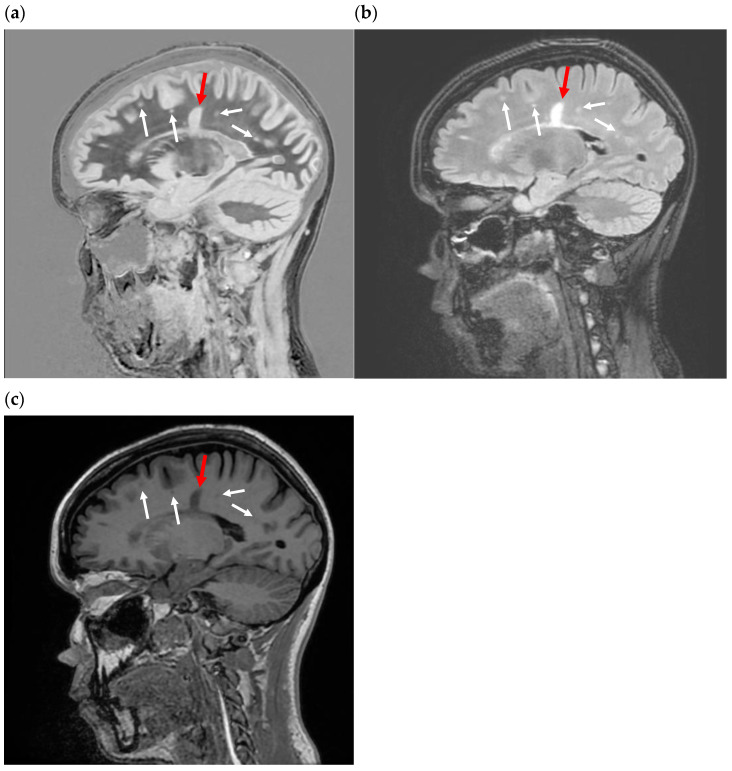
Sagittal subtracted inversion recovery (SIR) images in a patient with multiple sclerosis (MS). (**a**) Sagittal SIR image in an asymptomatic patient with MS presenting for a routine follow-up using a wide mD. TR = 5000. TI_short_ = 450 to null white matter. TI_long_ = 850 to null gray matter. This is considered a wide mD image. The normal white matter is black. A “Dawson’s finger” (red arrow) is seen as an increased signal. Small plaques (white arrows) are also identified as areas of increased signal. The increased signal is due to increased T_1_ in the abnormal white matter. The contrast on the image is described by the green curve in Figure 6a. (**b**) Sagittal T_2_-FLAIR image in the same patient as in (**a**). The “Dawson’s finger” and small plaques are also seen as areas of increased signal, only the signal is due to increases in white matter T_2_. All the plaques seen on the SIR were also seen on the T_2_-FLAIR. (**c**) Sagittal inversion recovery fast spin echo T_1_ image in the same patient as in (**a**,**b**). The Dawson’s finger and small plaques are dark compared to normal white matter, as per the curve shown in Figure 5a. This contrast is due to increases in white matter T_1_. Compare the contrast between normal and abnormal white matter with the image in (**a**). The abnormal white matter is more conspicuous in (**a**). This is because the maximum slope of the SIR filter (the green curve in Figure 6a) is nearly twice that of the IR filter (red and blue curves in Figure 6a and blue curve in Figure 5a). See Table 1.

**Figure 8 bioengineering-11-00441-f008:**
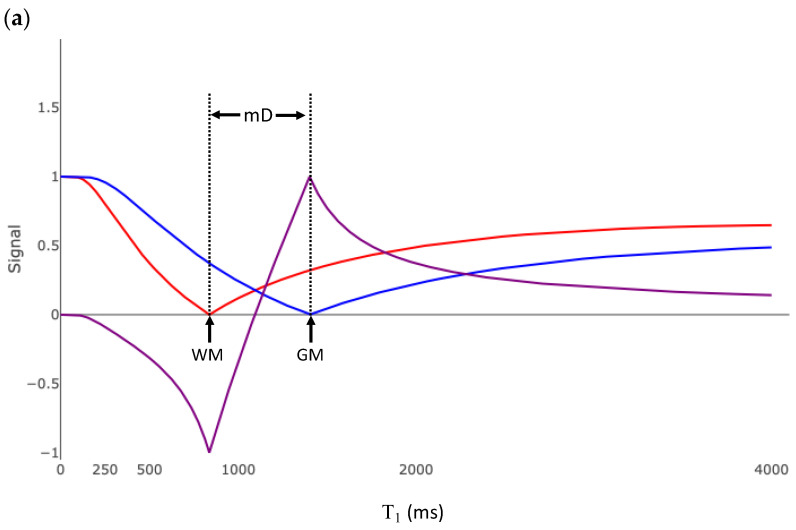
Divided Subtracted Inversion Recovery (dSIR) T_1_ filter. (**a**) The red curve is an inversion recovery (IR) sequence, with TI_short_ chosen to null white matter. The blue curve is the filter for an IR sequence, with TI_long_ chosen to null gray matter. The purple curve is the dSIR filter and is the division of the difference of the blue and red curves by their sum. The slope of the purple curve is 2.7 times the maximum slope of the curve in Figure 5a (which is the same as the maximum slope of the red and blue curves). The middle domain is the range of T_1_ values between the tissues nulled by TI_short_ and TI_long_, which in this case are white and gray matter. (WM—white matter; GM—gray matter; mD—middle domain). (**b**) T_1_ filter for the dSIR sequence with a narrow middle domain compared to the curve shown in (**a**). As the middle domain decreases, the slope of the purple curve increases. (**c**) T_1_ filter for a dSIR sequence targeted for changes in the T_1_ of gray matter. TI_short_ is chosen to null signal from gray matter. TI_long_ is chosen to be higher. The width of the middle domain determines the sensitivity of the sequence to small changes in T_1_. “WM” marks the T_1_ of white matter. “GM” marks the T_1_ of gray matter.

**Figure 9 bioengineering-11-00441-f009:**
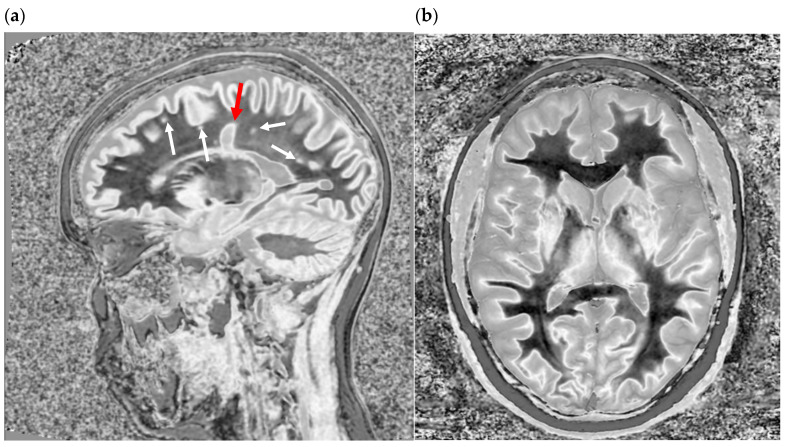
Divided Subtracted Inversion Recovery (dSIR) images. (**a**) Sagittal dSIR image of the same slice and patient as in Figure 7a–c. TR = 5000 ms. TI_short_ = 450 to null white matter. TI_long_ = 850 to null gray matter. TE = 7 ms. This is considered a wide mD image. The normal white matter is black. A “Dawson’s finger” (red arrow) is seen as an increased signal. Small plaques (white arrows) are also identified as areas of increased signal. The increased signal is due to increased T_1_ in the abnormal white matter. The contrast on the image is described by the purple curve in Figure 8a. The contrast between the normal and abnormal white matter is 2.7 times that of the SIR image in Figure 5 and 5 times that of the IR image in Figure 7c. (**b**) Axial narrow mD dSIR in a healthy volunteer. TR = 5000. TI_short_ = 350. TI_long_ = 500. TE = 7 ms, TR = 5000 ms. The normal white matter is black. Normal gray matter is intermediate signal. There is a high signal boundary between the gray and white matter, because the tissue filter (purple graph in Figure 8b) has a maximum between the T1 values of white matter and gray matter. Note that the gray matter is not as bright as on the wide mD dSIR image, as in (**a**). Compare the y-axis values (signal) of the purple curves in Figure 8a,b at the T_1_ of gray matter.

**Figure 10 bioengineering-11-00441-f010:**
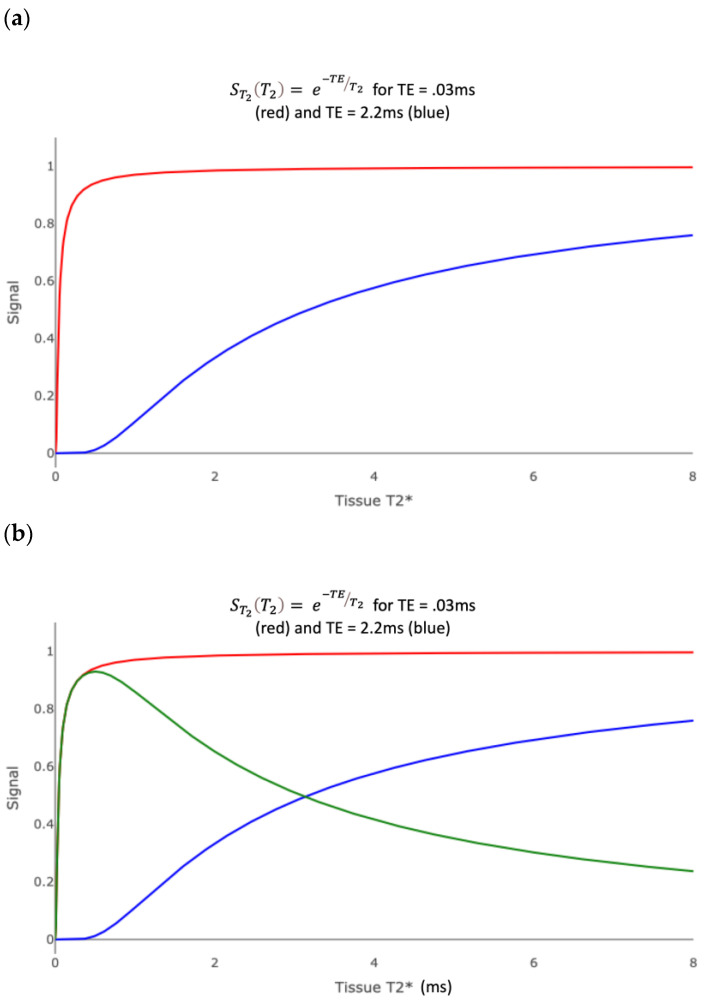
Divided echo subtraction (dES) T_2_* tissue filters. (**a**) T_2_* filter for ultrashort and short TE sequences. The red curve is for an ultrashort TE sequence with TE = 0.05 ms. The blue curve is for a short TE sequence with TE = 2.2 ms. (**b**) T_2_* filter for the echo subtraction (ES) sequence. The green curve is the difference of the red and blue curves from (**a**). The subtraction increases the contrast between ultrashort T_2_* tissues and short/normal T_2_* tissues. (**c**) T_2_* filter for divided echo subtraction (dES) sequence. The purple curve is the difference of the red and blue curves divided by their sum. The dES filter further increases the contrast between ultrashort T_2_* tissues and short/normal T_2_* tissues.

**Figure 11 bioengineering-11-00441-f011:**
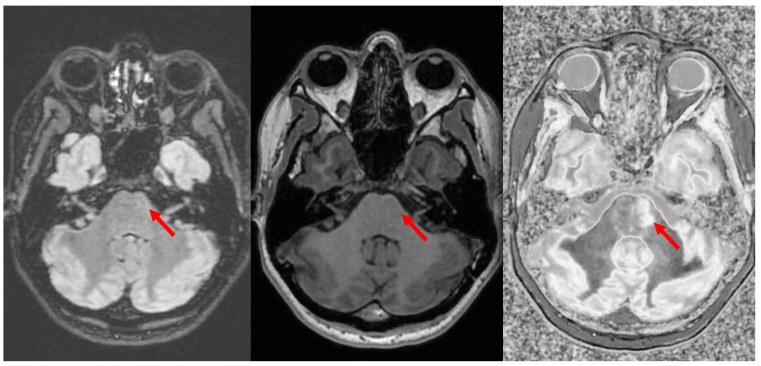
Divided Subtracted Inversion Recovery (dSIR) in a patient with multiple sclerosis (MS). T_2_-FLAIR (**left**), inversion recovery (IR) T_1_-weighted (**middle**), and wide-domain dSIR with TI_short_ = 450 ms and TI_long_ = 850 ms (**right**) images through the pons in a patient with MS. A large plaque is obviously present in the left hemipons on the dSIR image (red arrow in the image on the far right). The contrast in this image is due to changes in white matter T_1_. The change in T_1_ is insufficient to cause noticeable contrast on the IR T_1_ image (**middle**). The change in T_2_ is insufficient to cause noticeable contrast on the T_2_-FLAIR image (**left**).

**Figure 12 bioengineering-11-00441-f012:**
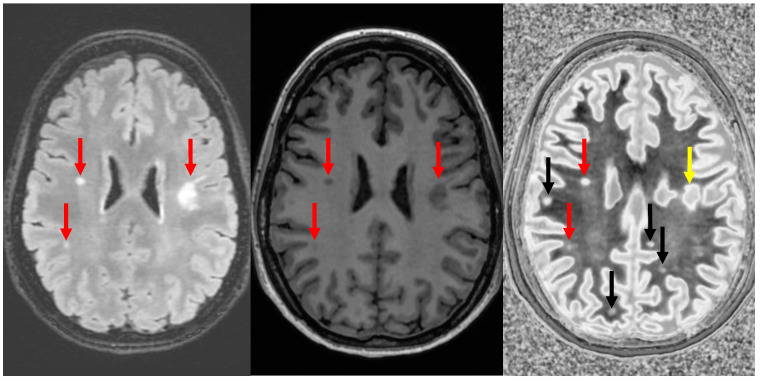
Divided Subtracted Inversion Recovery (dSIR) in a patient with multiple sclerosis (MS). T2-FLAIR (**left**), inversion recovery (IR) T_1_-weighted (**middle**), and wide-domain dSIR with TI_short_ = 450 ms and TI_long_ = 850 ms (**right**) images through the upper corona radiata in a patient with MS. Three plaques are seen on the T_2_-FLAIR and IR T_1_ images (red arrows). More plaques are seen on the dSIR image (black arrows). The plaque in the left frontal white matter is seen on the dSIR image (yellow arrow) but, due to the high signal etching along its margins, could easily be mistaken for cortex.

**Figure 13 bioengineering-11-00441-f013:**
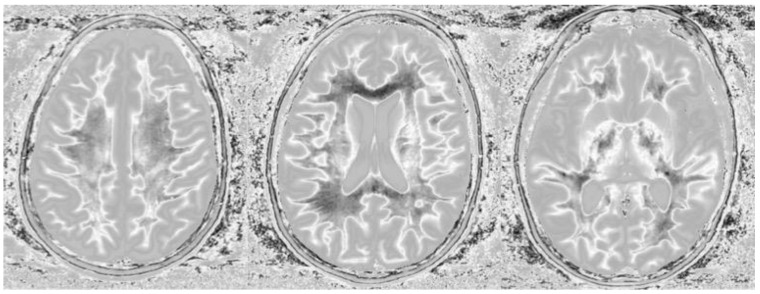
Divided Subtracted Inversion Recovery (dSIR) in a patient with multiple sclerosis (MS). Three axial narrow middle domain images in a patient with an acute MS flare at the level of the centrum semiovale (**left**), corona radiata (**middle**), and basal ganglia (**right**). TI_short_ = 350 ms. TI_long_ = 500 ms. TE = 7 ms, TR = 5000 ms. The white matter is not black as in Figure 9b. There is a widespread increased signal, though not a “white out” sign as described in Figure 14. This is an “intermediate” appearance but not considered normal.

**Figure 14 bioengineering-11-00441-f014:**
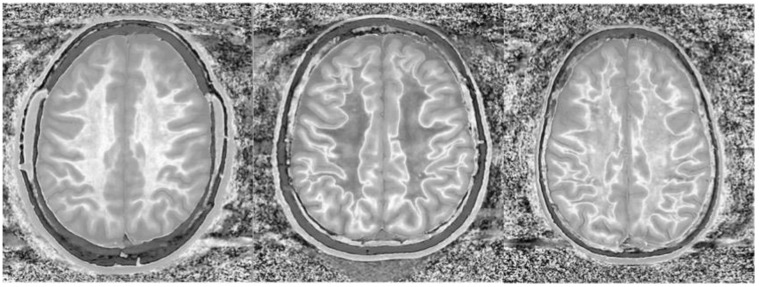
Normal and abnormal divided Subtracted Inversion Recovery (dSIR) images. Narrow middle domain images in three patients at the level of the centrum semiovale. TI_short_ = 350 ms. TI_long_ = 500 ms. TE = 7 ms, TR = 5000 ms. The left image shows an example of the “white out sign”, with a diffusely increased signal throughout the white matter. The center image is an example of normal. The white matter has a mildly increased signal that is normal because TI_short_ = 350 ms nulls tissue with T_1_ values less than that of white matter. The image on the right has an intermediate appearance, probably abnormal but not a “white out”.

**Figure 15 bioengineering-11-00441-f015:**
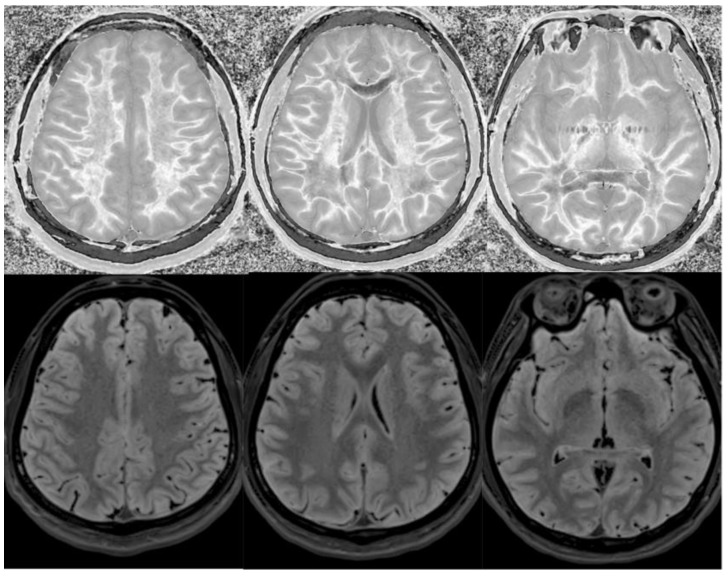
Divided Subtracted Inversion Recovery (dSIR) in a patient with Grinker’s myelinopathy. **Top row**: Narrow middle domain dSIR images at the level of the centrum semiovale (**left**), corona radiata (**middle**), and basal ganglia (**right**) in a patient with persistent symptoms following prolonged hypoxia due to a suicide attempt. TI_short_ = 350 ms. TI_long_ = 500 ms. TE = 7 ms, TR = 5000 ms. There is a diffuse “white out”. **Bottom row**: T_2_-FLAIR images at matching levels show normal-appearing white matter. Scans were obtained 9 months following injury.

**Figure 16 bioengineering-11-00441-f016:**
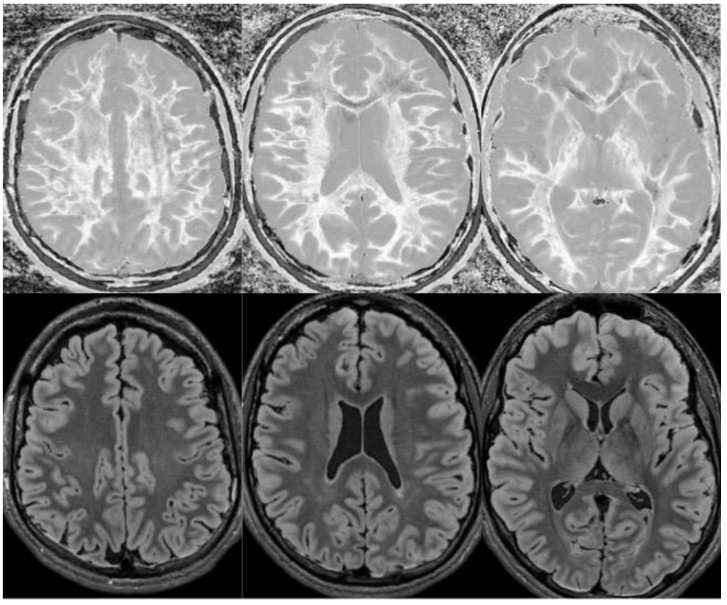
Divided Subtracted Inversion Recovery (dSIR) in a patient with Grinker’s myelinopathy. **Top row**: Narrow middle domain dSIR images at the level of the centrum semiovale (**left**), corona radiata (**middle**), and basal ganglia (**right**) in a patient with persistent symptoms following prolonged hypoxia due to drug overdose. TI_short_ = 350 ms. TI_long_ = 500 ms. TE = 7 ms, TR = 5000 ms. There is widespread “white out”, with some sparing in the deep frontal lobe white matter. **Bottom row**: T_2_-FLAIR images at matching levels show normal-appearing white matter. Scans were obtained 2 years following injury.

**Figure 17 bioengineering-11-00441-f017:**
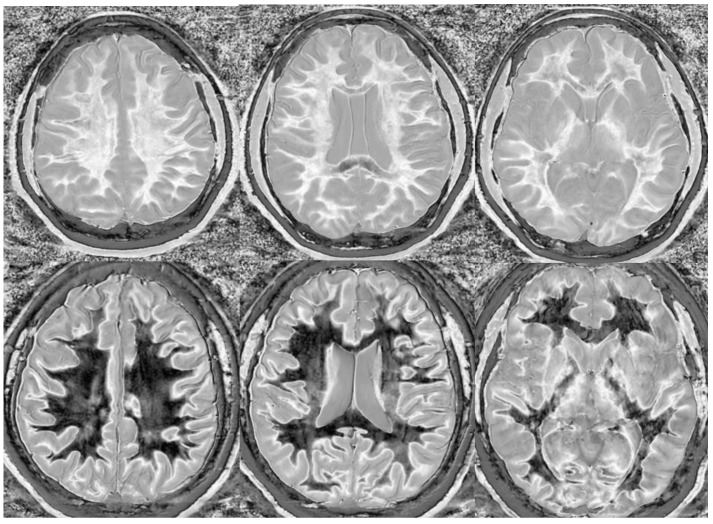
Two boys with mild head trauma. **Top row**: Narrow middle domain divided Subtracted Inversion Recovery (dSIR) images at the level of the centrum semiovale (**left**), corona radiata (**middle**), and basal ganglia (**right**) in two young men experiencing mild head trauma in the same rugby match. TI_short_ = 350 ms. TI_long_ = 500 ms. TE = 7 ms, TR = 5000 ms. Images were obtained within 5 days of injury. The player shown in the **top row** had symptoms of concussion at the time of imaging, and a “white out” sign is present. The player shown in the **bottom row** was asymptomatic, and the images appear normal.

**Figure 18 bioengineering-11-00441-f018:**
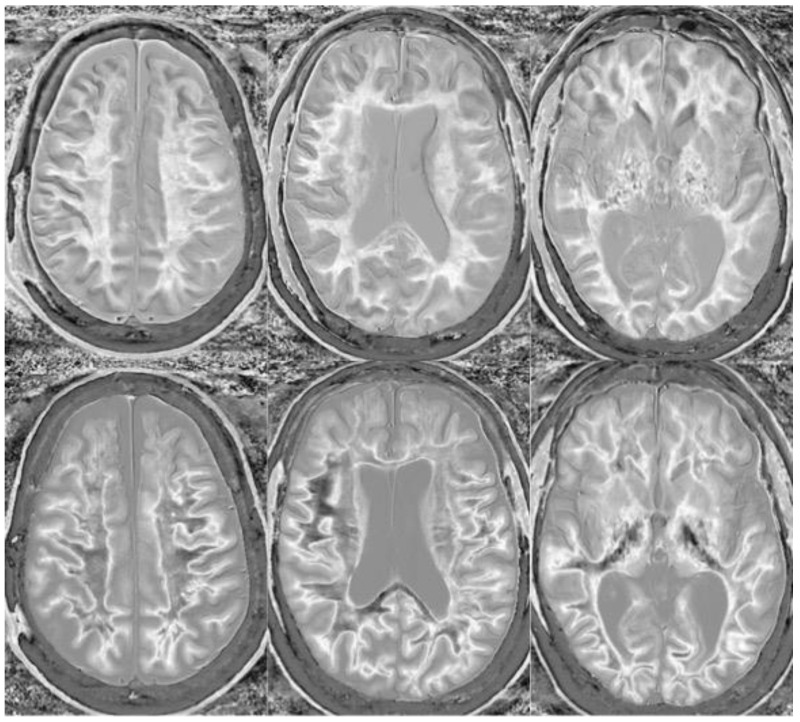
Divided Subtracted Inversion Recovery (dSIR) in a patient methamphetamine user. Narrow middle domain dSIR images at the level of the centrum semiovale (**left**), corona radiata (**middle**), and basal ganglia (**right**) in a volunteer immediately after a methamphetamine binge (**top row**) and 4 months into abstinence (**bottom row**) TI_short_ = 350 ms. TI_long_ = 500 ms. TE = 7 ms, TR = 5000 ms. The **top row** images show the “white out” sign, indicating diffuse mild white matter T_1_ elevation. The signal in the white matter partially normalizes on the bottom row. The appearance is closer to intermediate than normal, but there is definite improvement.

**Figure 19 bioengineering-11-00441-f019:**
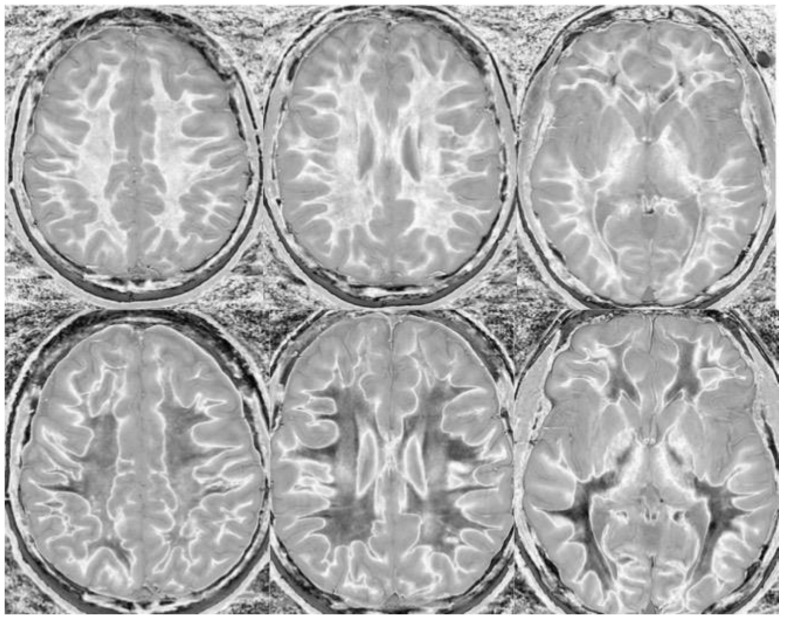
Divided Subtracted Inversion Recovery (dSIR) in a patient with a mild traumatic brain injury. Narrow middle domain dSIR images at the level of the centrum semiovale (**left**), corona radiata (**middle**), and basal ganglia (**right**) in a volunteer within five days of an mTBI (**top row**) and two weeks later (**bottom row**). TI_short_ = 350 ms. TI_long_ = 500 ms. TE = 7 ms, TR = 5000 ms. The **top row** images show the “white out” sign, indicating diffuse mild white matter T_1_ elevation. The signal in the white matter normalizes on the bottom row, where it appears normal.

**Figure 20 bioengineering-11-00441-f020:**
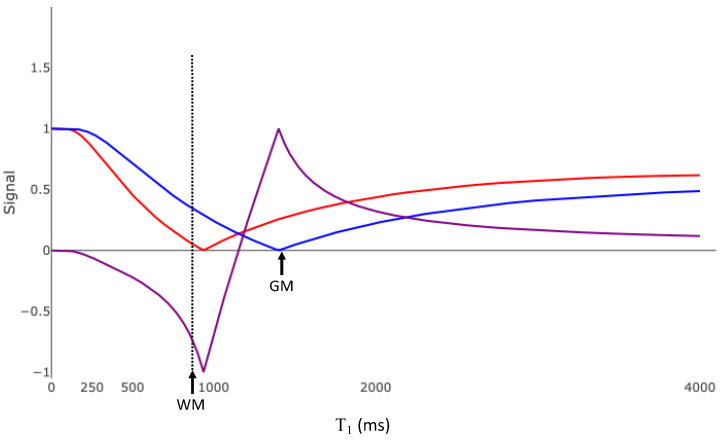
Divided Subtracted Inversion Recovery (dSIR) T_1_ filter with the shorter inversion time chosen too high. The red and blue curves are the T_1_ filters for IR sequences designed to null white (red curve) and gray (blue curve) matter. The TI for the red curve has been chosen too high. Small increases in T_1_ from normal in the white matter will result in decreased as opposed to increased signal on the dSIR filter (purple curve). GM—T_1_ value of gray matter. WM—T_1_ value of white matter.

**Figure 21 bioengineering-11-00441-f021:**
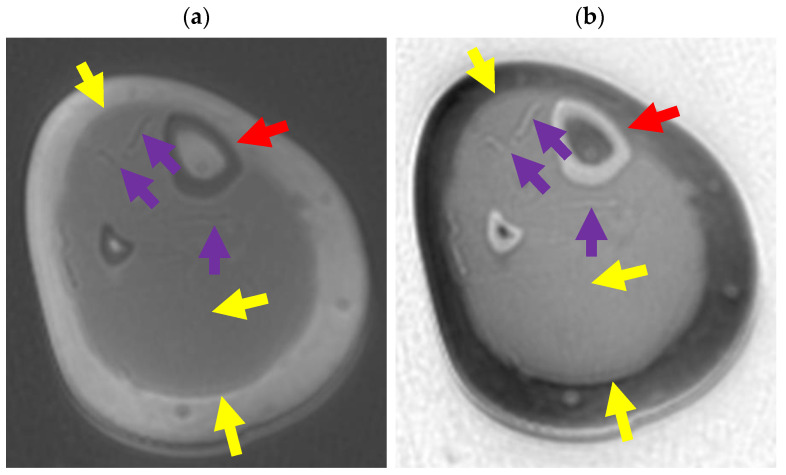
Divided echo subtraction (dES) (**a**) Axial ultrashort TE image through the lower leg with TE = 0.03 ms. Cortical bone (red arrows), and aponeuroses (purple arrows) are dark. The fascial layers (yellow arrows) are thin and poorly seen. (**b**) The same axial ultrashort TE image through the lower leg as in (**a**) displayed with an inverted gray scale. Cortical bone (red arrows) and aponeuroses (purple arrows) are bright. The fascial layers (yellow arrows) are thin and poorly seen. (**c**) Axial ES image created by subtracting a TE = 2.2 ms image from the TE = 0.03 ms image. The contrast between aponeuroses (purple arrows), fascia (yellow arrows), and muscle is increased compared to (**b**). (**d**) Axial dES image created by dividing the difference of the TE = 2.2 ms and TE = 0.03 ms images by their sum. The contrast between aponeuroses (purple arrows), fascia (yellow arrows), and muscle is improved compared to (**b**,**c**).

**Table 1 bioengineering-11-00441-t001:** Maximum slopes of the T1 tissue property filters.

Sequence	Relevant Parameters	Slope of the T_1_ Filter at the T_1_ of White Matter	Ratio of the Maximum Slope to the FSE Sequence
FSE	TR = 700 ms	−4.3 × 10^−4^/ms	1
FSE-IR	TR = 5000 ms, TI = 1000	8.2 × 10^−4^/ms	1.9
SIR	TR = 5000 ms, TI_short_ = 580, TI_long_ = 970	16 × 10^−4^/ms	3.7
dSIR wide mD	TR = 5000 ms, TI_short_ = 580, TI_long_ = 970	43 × 10^−4^/ms	10
dSIR narrow mD	TR = 5000 ms, TI_short_ = 580, TI_long_ = 740 ms	87 × 10^−4^/ms	20.2

FSE—fast spin echo; FSE-IR—fast spin echo inversion recovery; SIR—subtracted inversion recovery; sDIR—divided Subtracted Inversion Recovery; mD—middle domain; TR—repetition time; TE—echo time; TI_short_—shorter inversion time; TI_long_—longer inversion time.

## Data Availability

No formal data sets were created as part of this maniuscript.

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
