# Peer review of "Ultra-High Contrast MRI: Using Divided Subtracted Inversion Recovery (dSIR) and Divided Echo Subtraction (dES) Sequences to Study the Brain and Musculoskeletal System"

_bioengineering, 2024, doi:10.3390/bioengineering11050441_

Round 1

Reviewer 1 Report

Comments and Suggestions for Authors

1. Main Question:

The research explores the potential of divided subtraction MRI (dSIR and dES) as a novel technique to generate ultra-high contrast images for studying the brain and musculoskeletal system.

2. Originality and Relevance:

·  The paper introduces divided and subtracted MRI as a new processing method for generating high T1 or T2 contrast.

·  The application of dSIR for detecting subtle T1 changes in white matter, particularly in mTBI and other diffuse brain pathologies, is novel.

·  The potential of dSIR for targeted T1 contrast in various tissues (e.g., prostate cancer) is an interesting avenue for future exploration.

3. Contribution to the Field:

·  This study proposes a new method (dSIR and dES) for achieving superior T1 and T2 contrast in MRI examinations.

·  It highlights the potential of dSIR for identifying subtle white matter abnormalities in mTBI and other brain disorders, potentially improving diagnostic sensitivity.

·  The broader applicability of dSIR to target T1 changes in different tissues suggests its potential use in various disease settings.

4. Methodological Improvements and Controls:

·  The authors could consider including a more detailed description of the optimization process for sequence parameters in dSIR and dES.

·  Including validation studies with established diagnostic methods (e.g., histology) for mTBI and other applications would strengthen the evidence.

·  Further research with larger patient cohorts is necessary to establish the diagnostic accuracy of dSIR for mTBI and other conditions.

5. Consistency of Conclusions:

·  The paper claims dSIR offers 20 times higher sensitivity than traditional T1-weighted images for detecting subtle white matter changes. However, the evidence for this claim relies on the authors' research and would benefit from additional studies with objective comparisons.

·  The manuscript focuses on the potential of dSIR and dES, but a more balanced discussion acknowledging limitations (e.g., potential artifacts due to subtraction) would be valuable.

·  The study primarily explores dSIR in the context of white matter abnormalities. While future applications in other tissues are discussed, addressing these applications with specific experiments would strengthen the overall conclusions.

Overall, the study presents a promising new MRI processing technique with potential applications in brain and musculoskeletal imaging. Further research and validation are needed to fully establish its clinical utility.

Recommendations:

·  A more up-to-date reference list could be provided.

·  Benchmarks proving the effectiveness of the method could be expanded.

Reviewer 2 Report

Comments and Suggestions for Authors

The authors must correct the article in accordance with the comments.

This article addresses aspects of Ultra-High Contrast MRI in the task of studying the brain and musculoskeletal system.

The authors cover in detail aspects of the research performed.

However, there are a number of comments.

1. In all figures with graphs, starting from Figure 2, there is a problem with formulas in the headings of the figures: there are strange squares there.

2. Figure 6a needs to be made clearer. The font is very small.

3. In all figures with graphs, captions along the horizontal axis should be made immediately below it, and not as it is done now (along the lower edge of the figure).

4. It is necessary to add the Discussion section and compare the results obtained with those already known.

Comments on the Quality of English Language

 Minor editing of English language is required.

Round 2

Reviewer 2 Report

Comments and Suggestions for Authors

The article can be published in its presented form after finalizing the display of formulas in the figures.

Author Response

Dear Reviewer and Journal,

Thank you very much for the rapid turn around and positive reception of our manuscript.  I have replaced the figures that rendered incorrectly with .png images as opposed to pastings of powerpoint images.  As these equations previously rendered properly when I convert to a PDF I cannot confirm that the problem is solved.  Please let me know if this has fixed the problem on the journal's end.  If not then I will find another solution.  I did not replace any of the other figures in the manuscript.

Kind regards,

Daniel Cornfeld